# CD73-Mediated Immunosuppression Is Linked to a Specific Fibroblast Population That Paves the Way for New Therapy in Breast Cancer

**DOI:** 10.3390/cancers13235878

**Published:** 2021-11-23

**Authors:** Ilaria Magagna, Nicolas Gourdin, Yann Kieffer, Monika Licaj, Rana Mhaidly, Pascale Andre, Ariane Morel, Anne Vincent-Salomon, Carine Paturel, Fatima Mechta-Grigoriou

**Affiliations:** 1Equipe labellisée Ligue Nationale Contre le Cancer, Stress and Cancer Laboratory, Institut Curie, PSL Research University, 26, rue d’Ulm, 75005 Paris, France; ilaria.magagna@curie.fr (I.M.); yann.kieffer@curie.fr (Y.K.); monika.licaj@curie.fr (M.L.); rana.mhaidly@curie.fr (R.M.); 2Inserm, U830, 75005 Paris, France; 3Innate Pharma, 117 Avenue de Luminy BP 30191, 13276 Marseille, France; nicolas.gourdin@innate-pharma.fr (N.G.); pascale.andre@innate-pharma.fr (P.A.); ariane.morel@innate-pharma.fr (A.M.); carine.paturel@innate-pharma.fr (C.P.); 4Hospital Group, Department of Diagnostic and Theranostic Medicine, Institut Curie, 75005 Paris, France; anne.salomon@curie.fr

**Keywords:** breast cancer, CAF subsets, Tregs, immunosuppression, anti-CD73 antibody

## Abstract

**Simple Summary:**

Recent findings have revealed the contribution of cancer-associated fibroblasts (CAF) in immune escape in breast cancer. Still, how to specifically target immunosuppressive CAF remains an unmet medical question. Here, we provide a promising therapeutic strategy by highlighting the role of CD73 in immunosuppressive CAF. By studying cohorts of breast cancer patients and performing functional assays, our study uncovers how CD73 contributes to immunosuppression by acting in a specific CAF subpopulation (referred to as CAF-S1) in breast cancer. In addition, we validate that using an anti-CD73 antibody significantly reduces CAF-S1-mediated immunosuppression, thereby highlighting a new interesting therapeutic strategy for breast cancer patients.

**Abstract:**

Background: Cancer-associated fibroblasts (CAF) are heterogeneous with multiple functions in breast cancer. Recently, we identified a specific CAF subpopulation (referred to as CAF-S1), which promotes immunosuppression and immunotherapy resistance. Methods and Results: Here, by studying a large collection of human samples, we highlight the key function of CD73/NT5E in CAF-S1-mediated immunosuppression in breast cancer. We first reveal that CD73 protein level specifically accumulates in CAF-S1 in breast cancer patients. Interestingly, infiltration of regulatory T lymphocytes (Tregs) is significantly correlated with CD73 expression in stroma but not in epithelium, indicating that CD73 contributes to immunosuppression when expressed in CAF-S1 and not in tumor cells. By performing functional assays based on relevant systems using primary CAF-S1 isolated from patients, we demonstrate that CAF-S1 increase the content in both PD-1+ and CTLA-4+ Tregs. Importantly, the use of a blocking anti-CD73 antibody on CAF-S1 reduces CAF-S1-mediated immunosuppression by preventing expression of these immune checkpoints on Tregs. Conclusions: Our data support the potential clinical benefit of using both anti-CD73 and immune-checkpoint inhibitors in breast cancer patients for inhibiting CAF-S1-mediated immunosuppression and enhancing anti-tumor immune response.

## 1. Introduction

Breast cancers (BC) represent the most common female cancers worldwide and cause the greatest number of cancer-related deaths among women [1]. New treatments are necessary to improve BC patient outcome and survival. Exceptional advances have been achieved in cancer therapy with immune checkpoints inhibitors [2,3,4]. Immune checkpoints inhibitors represent a current therapy for some solid neoplasia, such as melanoma and lung cancer. Several clinical trials are currently on going to test Immunotherapies in BC patients. Nevertheless, only a subgroup of patients treated with these drugs shows a beneficial response [4], suggesting the existence of primary resistance mechanisms that favor escape from immune response. In this context, identification of new immunosuppressive mechanisms mediated by tumor cells or by cells of the tumor microenvironment (TME) might be extremely significant. Cancer-associated fibroblasts (CAF) are one of the most abundant TME components in solid tumors [5,6,7]. CAF are involved in multiple tumor-promoting activities, including tumor growth, metastatic spread, angiogenesis and resistance to chemotherapy [5,8,9,10,11,12,13,14]. The understanding of CAF heterogeneity in human cancer recently evolved. Indeed, recent studies identified different CAF subsets in several cancer types [13,15,16,17,18,19,20,21,22,23,24,25,26]. In breast and ovarian cancer, four distinct CAF subsets (named CAF-S1 to CAF-S4) have been recently identified by combining different fibroblastic markers, including fibroblast activation protein (FAP), Integrin β1 (CD29), smooth muscle α-actin (α-SMA) and platelet-derived growth factor receptor beta (PDGFRβ) [19,20]. CAF-S1 (FAP^+^ CD29^med-high^ α-SMA^med-high^ PDGFRβ^med-high^) and CAF-S4 (FAP^-^ CD29^high^ α-SMA^high^ PDGFRβ^low-med^) myofibroblastic subsets accumulate in aggressive BC subtypes, i.e., HER2 and triple-negative (TN) BC [19]. In line with these findings, CAF-S1 and CAF-S4 drive metastasis formation in axillary lymph nodes and late distant relapse in BC patients through complementary mechanisms [27,28]. Moreover, CAF-S1 fibroblasts promote immunosuppression by interacting with regulatory T lymphocytes (Tregs) in BC and ovarian tumors [19,20,26], thereby confirming previous observations in pancreatic cancer and in mouse models [13,24,25,29,30]. In particular, CAF-S1 increase the proportion of CD4+ CD25+ FOXP3+ T lymphocytes, enriched in Tregs, by promoting their attraction, survival and activation [19,20,26]. These data demonstrate that FAP+ fibroblasts (CAF-S1) could act as a source of resistance to anti-tumor immune response not only in mice [29] but also in humans, as recently demonstrated [24,25,26]. The mechanisms underlying CAF-S1-mediated immunosuppression and the way to target clinically these cells in order to circumvent their immunosuppressive activity remain open biological and medical questions.

The ecto-5′-nucleotidase CD73 (also referred as NT5E) is well-known to promote tumor immune escape through production of extracellular adenosine. CD73 converts extracellular AMP into extracellular adenosine (Ado) and works in tandem with CD39 (also named ectonucleoside triphosphate diphosphohydrolase 1, ENTPD1), responsible for the degradation of extracellular ATP and ADP into AMP [31]. In physiological conditions, Ado produced by CD73 enzymatic activity protects tissues against excessive inflammation. Indeed, Ado binds to purinergic adenosine receptors (A_2A_ and A_2B_) at the surface of lymphoid and myeloid cells, preventing their proliferation and effector functions [32]. CD73 is overexpressed in a wide range of human carcinomas, such as TN BC, and its expression is linked to poor prognosis [33,34,35,36,37]. As a rate-limiting enzyme in the generation of immunosuppressive Ado, CD73 represents an attractive target for cancer immunotherapy [38,39]. Interestingly, CD73 is highly expressed in FAP+ (CAF-S1) fibroblasts in breast, ovarian and colorectal cancer [19,20,40]. While CD73 expression and roles have been well established in cancer cells and in tumor-infiltrated lymphocytes [38,41], CD73 functions in CAF remain poorly investigated. Here, we address this question and show that high CD73 protein level, observed in CAF-S1-enriched BC patients, is associated with Tregs accumulation. In contrast, CD73 expression in cancer cells is not indicative of Tregs infiltration in BC, indicating that stromal CD73 is key in immunosuppression. In human primary CAF-S1 cell lines isolated from BC, we confirmed that CD73 is highly expressed and enzymatically active. Moreover, by performing functional in vitro assays using primary cells, we demonstrated that CAF-S1 fibroblasts increase the abundance of PD-1+, CTLA-4+ and PD-1+ CTLA-4+ double positive T cells among CD4+ CD25+ FOXP3+ Tregs. Interestingly, blockade of CAF-S1-dependent CD73 enzymatic activity by using a specific anti-CD73 antibody [39] impairs CAF-S1-mediated Tregs activation and immunosuppressive function, suggesting that BC patients might benefit from anti-CD73 immunotherapy. Taken as a whole, our data reinforce the potential clinical interest of using anti-CD73 antibody in clinical practice.

## 2. Materials and Methods

### 2.1. Description of Retrospective and Prospective Cohorts of Breast Cancer Patients

The projects developed here are based on surgical residues, available after histopathological analyses and not required for diagnosis. There is no interference with clinical practice. Samples from all patients included in this study were analyzed before patients received any treatment. Analysis of tumor samples was performed according to the relevant national law on the protection of people taking part in biomedical research. All patients included in our study were female (see #Institutional Review Board Statement and #Informed consent statement). HER2-amplified carcinomas have been defined according to ERBB2 immunostaining using ASCO’s guideline. Luminal (Lum) tumors were defined by positive immunostaining for ER (estrogen receptor) and/or PR (progesterone receptor). The cut-off used to define hormone receptor positivity was 10% of stained cells. LumA patients were defined according to Ki67 (proliferation) staining (Ki67 below 15%: LumA). TN immunophenotype was defined as follows: ER^-^PR^-^ERBB2^-^ with the expression of at least one of the following markers: KRT5/6^+^, EGF-R^+^, Kit^+^.

Retrospective cohort: The retrospective cohort was composed by *n* = 215 BC patients including LumA (*n* = 96), HER2 (*n* = 54) and TN (*n* = 65) and was analyzed by IHC. For this retrospective cohort, we had access to Tissue Micro-Arrays composed by 2 cores of BC surgical samples selected by Pathologist as representative for the whole tumor (diameters: 1 mm; thickness: 3 µm). The clinical features of BC patients are described in Table 1. To determine the CAF-subset enrichment in BC patients of each subtype, the staining of 6 different fibroblastic markers (FAP, CD29, α-SMA, PDDGFRβ, FSP-1, CAV) has been performed by immunohistochemistry (IHC) in LumA, HER2, and TN BC subtypes and the histological scores (Hscores) quantified, as described in [19,28]. The infiltration of CD3+ and FOXP3+ T cells has been quantified manually in the epithelial and stromal compartments, separately, considering 5 to 10 representative fields at 20× magnification per tumor and divided by the area of the section.

Prospective cohort: The prospective cohort was composed by *n* = 26 fresh BC samples prior treatment after the surgery and includes LumA (*n* = 12), HER2 (*n* = 3) and TN (*n* = 11) subtypes. Samples were collected and cultured in vitro to establish CAF primary cell lines (see below). The clinical features of prospective cohort are described in Table 1.

### 2.2. CD73 Immunohistochemistry (IHC) Staining in BC Samples

CD73 IHC staining in human BC samples was performed using formalin-fixed paraffin embedded (FFPE) tissues. CD73 IHC staining was performed in the Lab Vision IHC Autostrainer 480 (Thermo Scientific, Waltham, MA, USA). For the deparafination of FFPE samples, the standard protocol with xylene and alcohol gradient was used. After dewaxing procedure, FFPE samples were incubated in a solution citrate pH6 (Dako, Agilent Technologies, Santa Clara, CA, USA, #S236984-2) in the microwave for 20 min at 97 °C for antigen retrieval. The blockade of endogenous peroxidase was made with Dako REAL peroxidase-blocking solution (Dako, #S202386-2) for 10 min at room temperature (RT). Blocking of unspecific binding was performed using a Dako blocking serum-free solution ready-to-use (Dako, #X0909) for 10 min at RT. Sections were incubated with rabbit anti-human CD73 primary antibody (Sigma, Saint-Louis, MO, USA, #HPA017357, dilution 1:600) previously tested for its specificity or Rabbit IgG, polyclonal isotype control (abcam, #ab171870, dilution 1:1250) for 1 h at RT. The antibodies were diluted in Dako EnVision FLEX Wash Buffer (Dako, #K800721-2). After antigen detection, sections were incubated with HRP-conjugated secondary antibody goat anti-rabbit (ABC kit, Vector laboratories, Burlingame, CA, USA, #PK-6101) for 25 min at RT. The tissue sections were washed using Dako EnVision FLEX Wash Buffer (Dako, #K800721-2). Antibody detection was performed by incubation with avidin-horseradish peroxidase (Vectastain, Vector laboratories, Burlingame, CA, USA, #PK-6101) for 25 min at RT and detected with 3,3′-diaminobenzidine for 5 min at RT (Dako, #K3468). Counterstaining was performed with Mayer hematoxylin freshly prepared (Dako, #S3309). Tissue sections were then submitted to serial gradients of xylen prior and mounted with coverslip in an automatic device (Sakura, Flemingweg, The Netherlands, Tissue-Tek DRS).

### 2.3. Quantification of CD73 Histological Scores (HScores) in Epithelial and Stromal Compartments of BC Samples

In order to quantify the CD73 IHC staining in sections, the slides were scanned using Philips Ultra-Fast Scanner and visualized at high resolution in the Philips IMS 2.2 software for further analysis and photos. The CD73 staining was quantified as histological score (Hscore), calculated as percentage of stained cells (either cancer cells or fibroblasts) multiplied by the intensity of staining in epithelium and stromal compartments, separately. For each section, the percentages of stroma and epithelial compartments were based on morphology assessment. We referred as Epithelium the compartment composed by cancer cells which have round shape, large nucleus and low cytoplasm content. We considered Stroma the compartment composed by fibroblasts which have an elongated shape, tiny and elongated nucleus and scarce cytoplasm. The discrimination between the two compartments was verified by EPCAM (Epithelial cell adhesion molecule) and α-SMA staining performed by independent researcher for the previous studies [19,28].

### 2.4. Establishment of CAF-S1 Primary Cell Lines from BC Samples

To establish CAF-S1 primary cell lines for this study, we based on protocols described in previous studies [19,20,28]. In brief, BC samples were cut into small fragments (about 1 mm^2^) in sterile conditions and put in a previously grid-designed petri dish, in presence of DMEM (Hyclone, Cytiva, Logan, UT, USA, #SH30243.01) supplemented with 10% heat-inactivated fetal bovine serum (FBS) (Biosera Europe, Nuaille, France, #FB-1003/500) with streptomycin (100 µg/mL) and penicillin (100 U/mL) (Gibco, Thermo Scientific, Waltham, MA, USA, #15140122) (Supplemented DMEM). Tumor fragments were incubated at 37 °C, 1.5% O_2_ for at least 2 weeks. Primary CAF spread from small tumor fragments, adhered to plate and proliferated. When CAF reached 60–70% confluence, they were harvested with TrypLE (Gibco, #12605-10), washed and centrifuged (1200 rpm, 5 min) and put in a new petri dish with supplemented DMEM. CAF-S1 phenotype was assessed by flow cytometry, as described below. CAF-S1 were used until passage 10 to avoid senescence, commonly occurring with primary cell lines.

### 2.5. Characterization of CAF-S1 by Flow Cytometry Analysis

CAF-S1 phenotype was assessed by flow cytometry. CAF-S1 primary cell lines were used when they reached at least 90% confluence. CAF were harvested and suspended in PBS supplemented with 1% heat-inactivated human serum and 2 mM EDTA (PBS+) at 5 × 10^5^ cells in 50 µL and were moved in 96 well-plate V-bottom (Greiner, Kremsmünster, Austria, #651101) for the staining. Cells were stained with LIVE/DEAD™ fixable dye (Invitrogen, Thermo Fisher Scientific, Waltham, MA, USA, #L34957) (dilution 1:1000 in PBS) for 20 min at RT. After one washing with PBS and one centrifuge at 1500 rpm for 5 min, CAF were fixed with 4% PFA solution (PFA, Electron Microscopy Sciences, Philadelphia, PA, USA, #15710) for 20 min at RT. After one washing with PBS+, cells were stained with antibody or isotype control mixes for 45 min at RT. Antibodies and isotypes control were diluted in PBS+ supplemented with 0.1% saponin. The antibody mix contains anti-FAP-APC (primary antibody, 1:200, R&D Systems, Minneapolis, MN, USA, #MAB3715), anti-CD29-Alexa Fluor 700 (1:100, Biolegend, San Diego, CA, USA, #303020) anti-PDGFRβ-BV421 (1:50, BD Horizon, Franklin Lakes, NJ, USA, #564124) anti-SMA-Alexa^®^ Fluor 594 (1:25, R&D Systems, #IC1420T-025) antibodies. The isotype control mix contains purified mouse IgG1 isotype control (primary antibody, 1:200, R&D Systems, #MAB002), Alexa Fluor 700 mouse IgG1 isotype control (1:100, BioLegend, #400144), BV421 mouse IgG2A isotype control (1:100, BD Horizon, #562439) and Alexa^®^ Fluor 594 mouse IgG2A isotype control (1:25, R&D Systems, #IC003T). At the end of staining, cells were rapidly washed and suspended in PBS+. Stained CAF-S1 fibroblasts were analyzed using LSRFortessa^TM^ analyzer (BD biosciences). At least 5 × 10^5^ events were recorded. Compensations were performed using single staining on anti-mouse IgG and negative control beads (BD biosciences, #552843) for each antibody and ArC reactive beads (Molecular probes, Eugene, OR, USA, #A10346) for live/dead staining. Data were analyzed using FlowJo version 10.5.2.

### 2.6. Measurement of CD73 Protein Levels in CAF-S1 by Flow Cytometry

CD73 protein levels at the surface of CAF-S1 primary cells were assessed by flow cytometry. CAF-S1 were stained with unconjugated anti-CD73 Ab (Innate Pharma, Marseille, France) or the corresponding isotype control (Innate Pharma) for 20 min at RT protected from light. Antibody cloning, chimerization and purification were described in detail in a previous study [39]. After two serial washing with PBS+, CAF-S1 fibroblasts were stained with secondary antibody conjugated with PE fluorescent dye (1:200, Jackson Immuno research, West Baltimore Pike, Philadelphia, PA, USA, #109-116-160) for 20 min at RT. At the end of staining, cells were rapidly washed as in previous steps and were suspended in PBS+ for the analysis by flow cytometry. 2.5 µg/mL of DAPI were added just before the analysis. CD73 staining at the surface of CAF-S1 was analyzed using LSRFortessa^TM^ analyzer (BD biosciences). At least 5 × 10^5^ events were recorded. Data were examined using FlowJo 10.5.2. Alive and single cells were examined for CD73 protein level. The specific Median Fluorescence Intensity (SpeMFI) for CD73 was calculated as follow: MFI of CAF-S1 stained with anti-CD73 Ab/MFI of CAF-S1 stained with isotype control.

### 2.7. Measurement of Exogenous AMP Hydrolysis

CD73 enzymatic activity in CAF-S1 primary cell lines was assessed using AMP GLO^TM^ luminescence assay (Promega, Madison, WI, USA, #V5011) allowing the measurement of exogenous AMP hydrolysis. Harvested CAF-S1 were suspended in supplemented DMEM at the concentration of 4 × 10^6^ cells/mL. A total of 25 µL of CAF-S1 was plated in 96 well-plate U-bottom (Falcon, Corning, NY, USA, #353077) alone, in presence of anti-CD73 or isotype control antibody at 10 µg/mL, or the pharmacological inhibitor, APCP, at 100 µM. Cells were incubated 1 h at 4 °C. A total of 25 µL of AMP 50 µM diluted in supplemented DMEM was added for 1 h at 4 °C. After a rapid centrifuge, 25 µL of supernatants were then moved in 96-well white microplate (Grainer, #655083) to quantify the residual AMP with AMP-Glo^TM^ luminescence assay according to manufacturer’s instructions. Emitted light was measured on plate reader (Fluorstar Optima, BMG Labtech). The relative luminescence units (RLU) were proportional to the residual AMP within supernatants. The percentages of residual AMP were calculated as follow: %residual AMP = relative luminescence unit (CAF-S1 untreated or treated with anti-CD73, isotype CTL antibody or APCP)/relative luminescence unit (AMP alone) × 100. %degraded AMP = 100 − %residual AMP.

### 2.8. Isolation of Peripheral Mononuclear Cells (PBMCs) from Healthy Donors

PBMC were isolated from blood of healthy donors (HD) obtained from the ‘Etablissement Français du Sang (Paris, France). PBMC were isolated using density gradient centrifugation in presence of Lymphoprep (STEMCELL Technologies, Vancouver, BC, Canada #07861) previously cooled at 4 °C. PBMC were diluted in RPMI (Hyclone, #SH30027.01) supplemented with 10% heat-inactivated FBS with streptomycin (100 µg/mL) and penicillin (100 U/mL) (supplemented RPMI) at the concentration of 10^8^ cells/mL. Cell were stored at 4 °C overnight, before the isolation of CD4+ CD25+ T cells.

### 2.9. Isolation of CD4+ CD25+ T Cells from Healthy Donor PBMCs

A total of 5–7.5 × 10^8^ PBMC was used for CD4+ CD25+ T cells purification by Human CD4+ CD25+ Tregs Isolation kit (MiltenyiBiotec, Bergisch Gladbach, Germany. #130-091-301) according to manufacturer’s instructions. The purity of CD4+ CD25+ T cells was determined by flow cytometry. CD4+ CD25+ T lymphocytes were suspended in supplemented DMEM at 5 × 10^5^ cells in 50 µL for co-culture with CAF-S1.

### 2.10. Co-Culture of CAF-S1 with CD4+ CD25+ T Cells after anti-CD73 Antibody Treatment

To study the impact of CD73 blockade on the immunosuppressive function of primary CAF-S1, CAF-S1 were plated in 24-well plates at the concentration of 5 × 10^4^ cells in 50 µL. Cells were plated in supplemented DMEM containing the anti-CD73 antibody or the corresponding isotype control (Innate Pharma) at 10 µg/mL. CAF-S1 were incubated 24 h at 37 °C at 1.5% O_2_. The media were replaced by fresh supplemented DMEM just before addition of 5 × 10^5^ CD4+ CD25+ T lymphocytes to CAF-S1 alone or pre-treated with anti-CD73 or with isotype control. Co-cultures were incubated for 16 h at 37 °C, 20% O_2_. After incubation, T lymphocytes were collected and analyzed by flow cytometry. T cells were stained with LIVE/DEAD™ fixable dye (Invitrogen #L34968) for 20 min at RT. After washing with PBS+, T cells were suspended in 50 µL of antibody or isotype control mixes in PBS+ for 15 min at RT. The antibody mix was composed by anti-CD45-APC-Cy7 (1:20, BD Pharmigen, #557833), anti-CD3Alexa Fluor 700 (1:20, BD Biosciences, #557943), anti-CD4-BV650 (1:50, Biolegend 317436) anti-CD25-PE (1:20, Miltenyi Biotec, # 130-113-282), anti-PD-1-BV421 (1:20, BD Horizon, #562516) anti-CTLA-4-PE-Cy5 (1:10, BD Pharmigen, #555854), anti-CD39-PerCP-efluor 710 (1:100, Invitrogen, #46-0399-42) and anti-CD73-APC Ab (Innate Pharma). The isotype control mix was composed by the same anti-CD45-APC-Cy7 (1:20, BD Pharmigen, #557833), anti-CD3Alexa Fluor 700 (1:20, BD Biosciences, #557943), anti-CD4-BV650 (1:50, Biolegend 317436) Ab and PE mouse IgG2b isotype control (1:20, Miltenyi Biotec, 130-092-215), BV421 mouse IgG1 isotype control (1:20, BD Horizon, #562438), PE-Cy5 mouse IgG2a isotype control (1:10, BD Pharmigen, #555575), PerCP-e710 mouse IgG1 isotype (1:100, Invitrogen, #46-4714-82) and APC mouse IgG1 isotype control (1:80, Invitrogen, #17-4714-82). T cells were washed with PBS+ and centrifuged at 1800 rpm for 10 min at 4 °C. FOXP3 staining buffer set kit (eBioscience, #00-5523-00) was used to detect intra-nuclear FOXP3 protein. After fixation and permeabilization for 1 h at RT, T lymphocytes were incubated with anti-FOXP3-Alexa Fluor 488 (1:40, Thermofisher Scientific, #53-4776) or the corresponding isotype control (1:200, Thermofisher scientific, #53-4321-80) for 30 min at RT. Stained T cells were analyzed by LSRFortessa^TM^ analyzer (BD biosciences). At least 10^5^ events were recorded. Data were examined using FlowJo 10.5.2.

### 2.11. Treg Cell Suppressive Assay

This protocol was adapted from [42]. After CD4+ CD25+ T lymphocytes isolation, CD4+ CD25- T cells were kept overnight at 4 °C in TexMACS medium (Miltenyi Biotec, #130-097-196). CD4+ CD25+ T cells were stained with antibody mix for 15 min at RT. The antibody mix was diluted in PBS+ in presence of a pool of fluorescent-conjugated primary antibodies recognizing CD4-APC (1:20, Miltenyi Biotec, #130-113-210), CD25-PE-Cy7 (1:40, BD Pharmingen™, #557741) CD127-FITC (1:20, ThermoFisher Scientific, #11-1278-42), and CD45RA-PE (1:20, BD Pharmingen™, #555489) proteins. Alive CD4+ CD25^med^ CD127- CD45RA^low^ cells (regulatory T cells, Tregs) were sorted on MoFlo Astrios (Beckman Coulter). Tregs were centrifuged at 1200 rpm for 10 min and suspended in supplemented DMEM at the following concentration 5 × 10^5^ cells in 50 µL.

5 × 10^4^ CAF-S1 fibroblasts were cultivated 24 h before the sorting of Tregs at 37 °C, 1.5% O_2_ in 24 well-plate. Cells were suspended in medium containing or not anti-CD73 or anti-CD73 isotype control antibodies at 10 µg/mL. Media were replaced with 450 µL of supplemented DMEM just before Tregs were added. Tregs were co-cultured with CAF-S1 fibroblasts for 16 h at 37 °C, 20% O_2_ in 24-well plate. Upon 16 h of co-culture, Treg cells were recovered in 1.5 mL tubes and manually counted using Glasstic slides with 10 grids (Kova, #87144E). Tregs were resuspended in supplemented RPMI at 2 × 10^4^ cells in 50 µL.

Previously isolated CD4+ CD25- cells (Teff) were stained 15 min at 37 °C, 5% O_2_ with 5 µM CellTrace™ CFSE cell proliferation dye (Thermofisher, #C34554) at 10^6^ cells/mL in sterile pre-warmed PBS. After two washing, Teff were suspended in supplemented RPMI at 10^4^ cells in 100 µL. Suppression assay was performed in U-bottom 96 well plates (Falcon, #353077) during 4 days at 37 °C, 20% O_2_. CFSE-stained Teff cells (10^4^ cells/100 µL) were incubated with Treg cells (Treg: Teff ratio of 2:1) alone, Treg pre-incubated with CAF-S1 pre-treated or not with anti-CD73 or isotype control. Proliferation of effector T cells was stimulated with anti-CD3/CD28 beads (Gibco, #11131D, 10^3^ beads/100 µL). CSFE-stained Teff cells alone and CSFE-stained Teff cells in presence of anti-CD3/CD28 beads only were used as a negative and positive control of proliferation, respectively. After 4 days, T cells were collected in 15 mL tube and washed with PBS+. After discarding the supernatants, cells were resuspended in 150 µL of PBS+ and transferred in 96-well plate, V bottom. Cells were stained with anti-CD4 antibody (1:20, Miltenyi Biotec, #130-113-210) diluted in PBS+ for 15 min at RT protected from light. After a washing step, T lymphocytes were suspended in 50 µL of PBS+ and analyzed by LSRFORTESSA™ analyzer (BD biosciences). A total of 2.5 µg/mL of DAPI was added immediately previous the acquisition. At least 5 × 10^5^ events were recorded. Data were examined using FlowJo 10.5.2. Among alive and single cells, CD4+ CSFE+ T effector cells were selected. Among CD4+ FITC+ T cells, CFSE MFI was selected to assess the proliferation. According to negative control of proliferation, i.e., CD4+ CFSE+ T effector without anti-CD3/CD28 beads, MFI of CFSE of non-proliferating and MFI of CFSE of the whole CD4+ FITC+ population were assessed for each condition. The percentage of suppression of T effector cell proliferation has been calculated as follow: ((log2 (y) of Teff alone − log2(y) Teff + Treg)/(log2 (y) Teff alone)) × 100. Y corresponds to CFSE MFI in the whole population divided by CFSE MFI in the non-proliferating cells.

### 2.12. Statistical Analysis

As previously published by the “Stress and Cancer” lab headed by Dr. F. Mechta-Grigoriou [19,26,28], data shown are means ± SEM (unless otherwise specified) and indicated in each figure legend. The number of tumors analyzed, or the number of independent experiments performed are specified in each figure legend, with at least 3 independent experiments, unless otherwise specified. The statistical test types used are in agreement with data distribution and are indicated in figure legend. First, normality was checked using the Shapiro–Wilk test and then parametric or non-parametric two-tailed tests were applied accordingly. Differences were considered as statistically significant at values of *p* ≤ 0.05. Spearman’s correlation test was used to evaluate the correlation coefficient between two parameters. All statistical analyses were performed using Prism software.

## 3. Results

### 3.1. Accumulation of Regulatory T Lymphocytes in Breast Cancer Is Correlated with CD73 Expression in the Stroma

To investigate the link between CD73 expression and T lymphocyte infiltration in BC, we first characterized CD73 protein levels in a retrospective cohort of 215 BC patients, including the three molecular subtypes, i.e., Luminal (LumA), HER2 and triple-negative (TN) BC (Table 1 for detailed description of the retrospective cohort studied). In that aim, we had access to samples for which CAF subsets enrichment and T lymphocyte infiltration have been previously quantified [19]. CD73 protein level and pattern were assessed by immunohistochemistry (IHC) staining on these BC samples. We first observed that CD73 protein level was heterogeneous in the different BC subtypes, as well as from one tumor to another within the same BC subtype (Figure 1A). We quantified CD73 protein levels by assessing CD73 histological scores (HScores) in each BC sample. When we first quantified CD73 Hscores in total BC sections without differentiating stromal and epithelial staining, we did not see any difference between BC subtypes (Appendix A), thereby confirming CD73 RNA levels found in bulk BC of the TCGA cohort (Appendix A). However, different CD73 protein levels were observed not only between different BC subtypes, but also between cellular compartments. Some tumors showed a strong staining in cancer cells (epithelial compartment) but a weak staining in CAF (stromal compartment), while other tumors exhibited high staining in CAF but low in cancer cells. We thus evaluated CD73 protein levels in each BC sample considering epithelial and stromal compartments separately (Figure 1B). By this way, we observed that CD73 protein level was higher in the epithelium of LumA (median CD73 Hscore = 294) compared to HER2 (median Hscore = 260) and TN BC (median Hscore = 237) (Figure 1B, left). In contrast, CD73 protein level was higher in the stroma of HER2 (median Hscore = 100) than in LumA (median Hscore = 56) and TN (median CD73 Hscore = 69) BC (Figure 1B, right). Considering this heterogeneity between epithelial and stromal CD73 histological scores, we next evaluated if there was any association between CD73 protein levels in epithelium and stroma in each BC subtype. Notably, we found a positive correlation between CD73 staining in stroma and epithelium in LumA and TN, but not in HER2 BC (Figure 1C). CD73 was shown to be highly expressed at RNA levels in a specific CAF subpopulation expressing FAP (FAP+ CAF-S1 fibroblasts) [19,20]. We next investigated CD73 protein levels in the different CAF subpopulations we previously identified in BC [19]. To do so, we took advantage of the CAF-subset enrichments that we have established in the retrospective cohort of BC patients studied here [19] (see also Table 1). Most LumA tumors were enriched in CAF-S2 and CAF-S4 subsets, HER2 tumors in CAF-S4 or at a lesser extent in CAF-S1, and TN tumors either in CAF-S1 or in CAF-S4 (Table 1). Remarkably, in all BC subtypes, CD73 protein level was significantly higher in the stroma of CAF-S1-enriched tumors (median CD73 Hscore = 169) compared to CAF-S2- (median Hscore = 32.5) or CAF-S4-enriched (median Hscore = 73) BC (Figure 1D), thereby showing that CD73 accumulates in CAF-S1 fibroblasts in BC, as observed at mRNA levels [19] (Appendix A).

As CAF-S1 fibroblasts are able to promote FOXP3+ T lymphocyte accumulation and activity [19,20], we next evaluated the potential association between CD73, as a single marker of immunosuppressive CAF-S1 fibroblasts, and FOXP3+ T lymphocytes enriched in regulatory T cells (Tregs). As previously reported [19], the content in both CD3+ and FOXP3+ T lymphocytes was higher in TN (median FOXP3+ cells/mm^2^ = 22.9) and HER2 (median FOXP3+ cells/mm^2^ = 23.8) BC than in LumA (median FOXP3+ cells/mm^2^ = 7.7) BC (Appendix A). Interestingly, FOXP3+ T lymphocyte infiltration was significantly correlated with CD73 protein levels in the stroma but not in the epithelium of LumA and TN BC (Figure 1E,F), suggesting that CD73-mediated immunosuppression is exerted through CD73 expression in CAF rather than in cancer cells. In contrast, neither CD73 protein level in the stroma nor in the epithelium was associated with global CD3+ T lymphocyte infiltration (Figure 1G,H), showing that stromal CD73 is associated with FOXP3+-enriched micro-environment, but not with a global T cell infiltration. Taken as a whole, these data show that CD73 protein accumulates both in stroma and in epithelium in a coordinated but different manner. Interestingly, CD73-stromal expression is significantly associated with FOXP3+ T cell content, while CD73 expression in epithelium is not, thereby suggesting that CD73 protein in the stroma may drive its immunosuppressive function.

### 3.2. CAF-S1 Primary Fibroblasts Exhibit High CD73 Enzymatic Activity

Considering the positive correlation between stromal CD73 protein and FOXP3+ T cell content in BC, we next defined the specific function of CAF-S1-expressed CD73 on FOXP3+ T lymphocytes by performing functional assays. To do so, we isolated CAF-S1 primary cell lines from human BC surgical specimens (Appendix A) and confirmed the CAF-S1 identity by multicolor flow cytometry combining several fibroblast markers (FAP, CD29, α-SMA, PDGFRβ) (Figure 2A,B), as previously shown in [19,20,26]. 18 human primary CAF-S1 cell lines were successfully isolated from fresh samples from 18 different BC patients of the 3 molecular subtypes (LumA, HER2, TN BC) and their CD73 surface protein level was assessed by flow cytometry. We confirmed that CD73 protein was detected at the surface of all CAF-S1 primary cell lines but at variable levels, as observed in stroma of BC patients (Specific CD73 MFI: Mean = 56; SEM = 11; Min = 14, Max = 216) (Figure 2C,D). We next showed that CD73 protein was enzymatically active in CAF-S1 primary cell lines through the measurement of exogenous AMP hydrolysis (Figure 2E). Interestingly, CD73 enzymatic activity was significantly correlated with its protein level at the surface of CAF-S1 fibroblast cell lines (Figure 2F). These data show that CAF-S1 cell lines isolated from human BC express active CD73 enzyme at their surface, thereby confirming that these established cell lines constitute appropriate in vitro cellular models to study stromal CD73 function.

### 3.3. CD73+ CAF-S1 Primary Fibroblasts Promote the Increase in PD-1+ CTLA-4+ FOXP3+ CD4+ CD25+ T Lymphocytes

As we showed that CD73 protein level in stroma is correlated with FOXP3+ T cell content in BC patients (Figure 1), we investigated the role of CAF-S1-expressed CD73 on CD4+ CD25+ T lymphocytes. We first performed functional assays by co-culturing primary CAF-S1 fibroblasts isolated from BC patients with CD4+ CD25+ T lymphocytes from healthy donors (Appendix A). As reported in breast and ovarian cancers [19,20,26], we confirmed that primary CAF-S1 cells co-cultured with CD4+ CD25+ T cells significantly increased the percentage of CD4+ CD25+ FOXP3+ T lymphocytes (Fold-change = 1.6) (Figure 3A,B), with a strong impact on CD4+ CD25+ FOXP3^high^ sub-population (Fold-change = 4.2) (Figure 3B, right). In addition to the increase in the percentage of FOXP3+ CD4+ CD25+ T lymphocytes, CAF-S1 also stimulated FOXP3 protein level in these cells (Fold-change = 1.2) (Figure 3C). Interestingly, we found that CAF-S1 fibroblasts significantly enhanced the percentages of PD-1+ (Figure 3D,E) and CTLA-4+ cells (Fold-changes = 1.2 and 1.4) (Figure 3G,H) among CD4+ CD25+ FOXP3+ lymphocytes, as well as among CD4+ CD25+ FOXP3^low-med^ and CD4+ CD25+ FOXP3^high^ T cells. Moreover, CAF-S1 significantly increased CTLA-4 protein level at the surface of CTLA-4+ FOXP3+ T lymphocytes (Fold-change = 1.3), but not PD-1 protein level at the surface of PD-1+ FOXP3+ T cells (Figure 3F,I). Furthermore, CAF-S1 cell lines significantly increased the percentages of PD-1+ CTLA-4+ double-positive cells among CD4+ CD25+ FOXP3+ T lymphocytes (Fold-change = 1.5) (Figure 3J,K). On the opposite, the proportion of double-negative PD-1- CTLA-4- cells among CD4+ CD25+ FOXP3+ T lymphocytes was reduced in presence of CAF-S1 primary cell lines (Fold-change = 0.8) (Figure 3J,K). In contrast to PD-1 and CTLA-4, CD73 or CD39 protein levels and percentages of positive cells among CD25+ FOXP3+ Tregs did not significantly change in presence of CAF-S1 fibroblasts (Figure 3L–N and Appendix A). In conclusion, CAF-S1 fibroblasts significantly increase the content in PD-1+ and CTLA-4+ T cells among Tregs, thereby highlighting the key function of CAF-S1 in regulating checkpoint inhibitors at the surface of T lymphocytes.

### 3.4. Anti-CD73 Antibody Impairs CAF-S1-Mediated Immunosuppression by Preventing PD-1 and CTLA-4 Up-Regulation in Tregs

As CAF-S1 fibroblasts exhibited high CD73 protein levels and enzymatic activity, we next tested the impact of CD73 inhibition on CAF-S1-mediated immunosuppression. To do so, we evaluated the effect of a humanized anti-CD73 monoclonal antibody on human CAF-S1 fibroblasts [39]. This antibody has been shown to specifically recognize CD73 protein and to efficiently inhibit its enzymatic activity [39]. We first evaluated the capacity of the anti-CD73 antibody (Ab) to inhibit CAF-S1 CD73 enzymatic activity by testing several antibody concentrations. The anti-CD73 Ab inhibited CD73 enzymatic activity of CAF-S1 in a dose dependent manner (Figure 4A). In parallel, we compared the efficacy of the anti-CD73 Ab to a chemical CD73 inhibitor, the Adenosine 5′-(α, β-methylene) diphosphate (APCP) (Figure 4B) and to an isotype control (IC) antibody (Figure 4C). We found that the impact of anti-CD73 Ab—at the chosen concentration for further analysis (10 μg/mL)—was as efficient as APCP (Figure 4B), reducing the CD73 enzymatic activity by an average of 62.5%, while the IC control had not effect, as expected (Figure 4C).

We next evaluated the impact of anti-CD73 Ab on CAF-S1-mediated immunosuppressive activity and compared it to IC (Figure 4D–O). Pre-incubation of CAF-S1 with anti-CD73 Ab significantly prevented the increase of CD4+ CD25+ FOXP3+ T lymphocyte content by CAF-S1 (Figure 4D,E left). This effect was observed for both CD4+ CD25+ FOXP3^high^ and FOXP3^low-med^ T cells (Figure 4D,E middle, right). In addition, CD73 inhibition on CAF-S1 also prevented CAF-S1-mediated increase of FOXP3 protein level at the surface of FOXP3+ CD4+ CD25+ T lymphocytes (Figure 4F). Interestingly, blockade of CD73 on CAF-S1 also prevented the increase of PD-1+ (Figure 4G,H) and CTLA-4+ T cell percentages among CD4+ CD25+ FOXP3+ lymphocytes (Figure 4J,K), in particular among CD4+ CD25+ FOXP3^high^ T cells (Figure 4H,K right). Moreover, CAF-S1 pre-treatment with anti-CD73 Ab impaired the increase of CTLA-4 protein levels at the surface of CTLA-4+ CD4+ CD25+ T lymphocytes (Figure 4L). The increase of PD-1+ CTLA-4+ double positive cells among CD4+ CD25+ FOXP3+ lymphocytes by CAF-S1 was also affected by pre-treatment of CAF-S1 fibroblasts with anti-CD73 Ab (Figure 4M,N left). Thus, all immunosuppressive features induced by CAF-S1 on CD4+ CD25+ T lymphocytes are abrogated by inhibition of CD73 activity in CAF-S1 fibroblasts.

Based on the impact of the anti-CD73 Ab on CAF-S1-induced Tregs features, we next investigated if CD73 blockade on CAF-S1 could also impair Treg activity, i.e., their ability to suppress effector T lymphocyte proliferation. We previously showed that CAF-S1 fibroblasts enhanced the ability of Tregs to suppress the proliferation of effector T cells [19,20]. Interestingly, CD73 inhibition on CAF-S1 prevented CAF-S1 to enhance Tregs activity, meaning Tregs capacity to reduce effector T cell proliferation (Figure 4O,P). Thus, the anti-CD73 Ab not only decreased CAF-S1 capacity to modulate Treg characteristics but also immunosuppressive function. Taken as a whole, these data show that CD73 is a key actor of CAF-S1-mediated immunosuppression. Consistent with this observation, CAF-S1-immunosuppressive activity is efficiently reduced by the use of anti-CD73 antibody.

## 4. Discussion

An important contributing factor to the poor clinical outcomes in BC is the highly immunosuppressive TME. This observation highlights the need to identify new mechanisms promoting tumor immune escape that can be targeted for increasing immunotherapy efficiency. The contribution of CAF in the promotion of immunosuppressive TME has been previously described [7,12,43,44,45]. Several mechanisms related to CAF-mediated immunosuppression have been reported. Many of them include secretion of different soluble factors (TGF-β, IL-6, CXCL12, and CCL2), which promote recruitment and activation of immunosuppressive populations, especially Tregs [5,9,17,19,20,29,30,46,47]. A meta-analysis has shown that FOXP3+ Tregs have a significant negative effect on overall survival (OS) in BC patients, and are associated with estrogen and progesterone receptor status, as well as with lymph node metastases [48,49,50]. In line with these data, we observed here that the global infiltration of CD3+ T cells, in particular of FOXP3+ T lymphocytes, is higher in TN BC than in Lum BC, thereby confirming previous studies [19,51,52,53]. Despite high T cell infiltration, recent data from clinical trials demonstrated that there is high proportion of TN BC patients who do not respond to immunotherapies [54]. Importantly, CAF-S1 enrichment is associated with increased Tregs infiltration in TN BC and with immunotherapy resistance in melanoma and lung cancer patients [19,26]. Interestingly, CD73, an enzyme that promotes tumor immune escape through production of extracellular adenosine, is highly expressed in CAF-S1. Moreover, we demonstrate here that CD73 is a key actor in CAF-S1-mediated immunosuppression, thereby arguing that CD73 might be an interesting therapeutic target for improving immunotherapy sensitivity.

During the last decade, numerous evidences revealed the role of CD73 as alternative immune checkpoint, which prevents anti-tumor immune response [38,55,56,57]. Indeed, as CD73 catalyzes the production of immunosuppressive adenosine, it has been hypothesized that CD73 prevents tumor destruction by inhibiting antitumor immunity. Several studies demonstrate that CD73 expressed on host hematopoietic and tumor cells impairs anti-tumor immune escape in colon, prostate and breast cancer mouse models [41,47,58,59,60,61]. However, the role of CD73 in CAF within human cancers remained poorly described.

In the present study, we go a step further by highlighting the predominant function of CD73 in CAF on Tregs infiltration in human BC. Indeed, we provide here one of the first in-depth and highly-resolutive study on CD73 protein function in human breast cancer. Consistent with our findings, CAF were also recently identified as a major source of CD73 to have an impact on colorectal cancer development [40]. Indeed, abundance of CD73+ CAF is associated with elevated CD73 activity and poor prognosis in colorectal cancer [40]. This is consistent with the high CD73 protein level we reveal here in the stroma of TN and HER2 BC, the two most aggressive BC subtypes. CD73 expression on T lymphocytes is stimulated by TGFβ-signaling pathway [62]. Interestingly, we previously highlighted the role of TGFβ signaling in CAF-S1 functions and its reciprocal crosstalk with cancer cells [28], suggesting TGFβ might be a key regulator of CD73 expression in CAF-S1. Moreover, the crosstalk between CAF and T lymphocytes plays an important role in the regulation of stromal CD73 expression. Indeed, ex vivo study with CAF isolated from lung cancer showed how T cells up-regulates CD73 on CAF in lung cancer through IFNγ secretion [63].

In addition, we highlight that CD73 is highly expressed in one particular CAF subset, characterized by high level of FAP and referred to as CAF-S1, and that CD73 expression in these cells promotes the increase in the proportion of CD4+ CD25+ FOXP3+, PD-1+ and CTLA-4+ T lymphocytes. Moreover, we demonstrate that CD73-specific antibody impairs CAF-S1-mediated immunosuppression, thereby highlighting that CAF-S1 fibroblasts exert their immunosuppressive activity in a CD73-dependent manner. Low CD73 expression and high global infiltration of CD45+ immune cells exhibit additive prognostic values and are indicative of a good clinical outcome in TN BC patients with invaded lymph nodes [64]. In agreement with these data and in line with the correlation between stromal CD73 expression and Treg infiltration we report here in LumA and TN BC, high content in cytotoxic CD8+ T lymphocytes (assessed by the CD8+/CD3+ ratio) is associated with a good overall survival in BC patients [19]. Consistent with these observations, pre-clinical studies revealed that CD73 expression inhibits specific anti-tumor CD8+ T cell response in mammary tumors [58]. Moreover, bone marrow transplantations established that both hematopoietic and non-hematopoietic CD73 expression are important to promote tumor immune escape in these mouse models [58].

Thus, while CD73 has been initially defined as a co-signaling molecule on T lymphocytes, it is now clear that CD73 exerts pro-tumorigenic functions in tumor cells but also in stromal fibroblasts, as we highlight here on Treg accumulation. Stromal CD73 represents a potential novel immune checkpoint in BC that can be effectively targeted in CAF using anti-CD73 antibody. In addition, our data support the interest to combine anti-CD73 therapy with anti-CTLA-4 and/or anti-PD-1 to overcome potential resistances related to concomitant PD-1 and CTLA-4 up-regulation in FOXP3+ T lymphocytes.

## 5. Conclusions

Cancer-associated fibroblasts are the most dominant cell type in TME of breast tumors. Numerous findings have convincingly demonstrated the tumor promoting effects of CAFs in breast tumors, including their contribution to immunosuppression. Hence, CAF represents an emergent interesting target of immunotherapy. This study reveals that CD73-depended immunosuppression mediated by specific CAF subset (CAF-S1) in breast cancer can be impaired using a blocking anti-CD73 monoclonal antibody. These results may open new therapeutic perspectives on the use of anti-CD73 Ab as compound for novel strategy to target breast CAF.

## Figures and Tables

**Figure 1 cancers-13-05878-f001:**
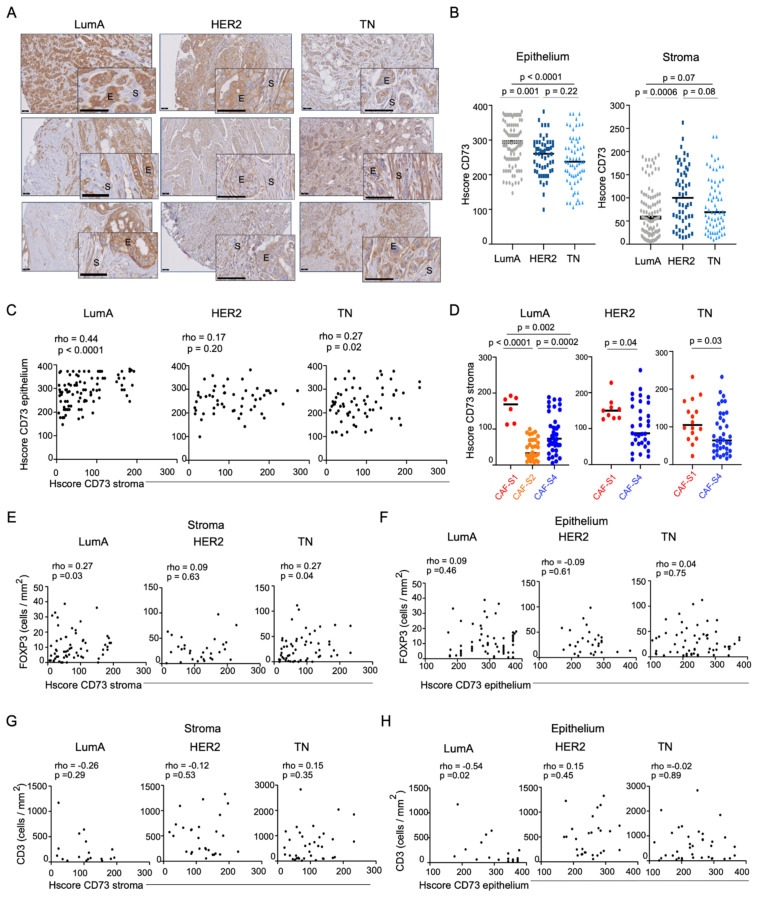
CD73 protein level in stroma is associated with Tregs content. (**A**) Representative images of CD73 immunostaining in LumA (**left**), HER2 (**middle**), TNBC (**right**) in stroma (S) and epithelium (E). Three representative patients per BC subtype are shown. Scale bars, 50 µm; 25 µm (inset). (**B**) Histological scores (HScore) of CD73 assessed in epithelial cells (epithelium, **left**) or CAF (stroma, **right**). Each dot represents one tumor (*n* = 215), including LumA (*n* = 96), HER2 (*n* = 54) and TN (*n* = 65) BC. Hscore is calculated as follow: HScore = Intensity of the marker (0–4) × % of stained cells (0–100). Data are represented as median. *p*-values from Mann–Whitney test. (**C**) Correlation between CD73 Hscores in stroma and epithelium in LumA (*n* = 96), HER2 (*n* = 54) and TN (*n* = 65) BC. Correlation coefficients and *p*-values from Spearman’s test. (**D**) CD73 HScore in stroma in CAF-S1-, CAF-S2- or CAF-S4-enriched BC (as determined in [19]) in LumA (*n* = 80), HER2 (*n* = 42) and TN (*n* = 53) BC. Data are represented as median. *p*-values from Mann-Whitney test for LumA and TN BC, and unpaired *t*-test for HER2 BC. (**E**,**F**) Correlation between FOXP3+ T lymphocytes per mm^2^ and HScore of CD73 in stroma (**E**) and epithelium (**F**) of LumA (*n* = 63), HER2 (*n* = 30), and TN (*n* = 57) BC. Correlation coefficients and *p*-values from Spearman’s test. (**G**,**H**) Correlation between CD3+ T lymphocytes per mm^2^ and HScore of CD73 in stroma (**G**) and epithelium (**H**) of LumA (*n* = 18), HER2 (*n* = 28) and TNBC (*n* = 36) patients. Correlation coefficients and *p*-values from Spearman’s test.

**Figure 2 cancers-13-05878-f002:**
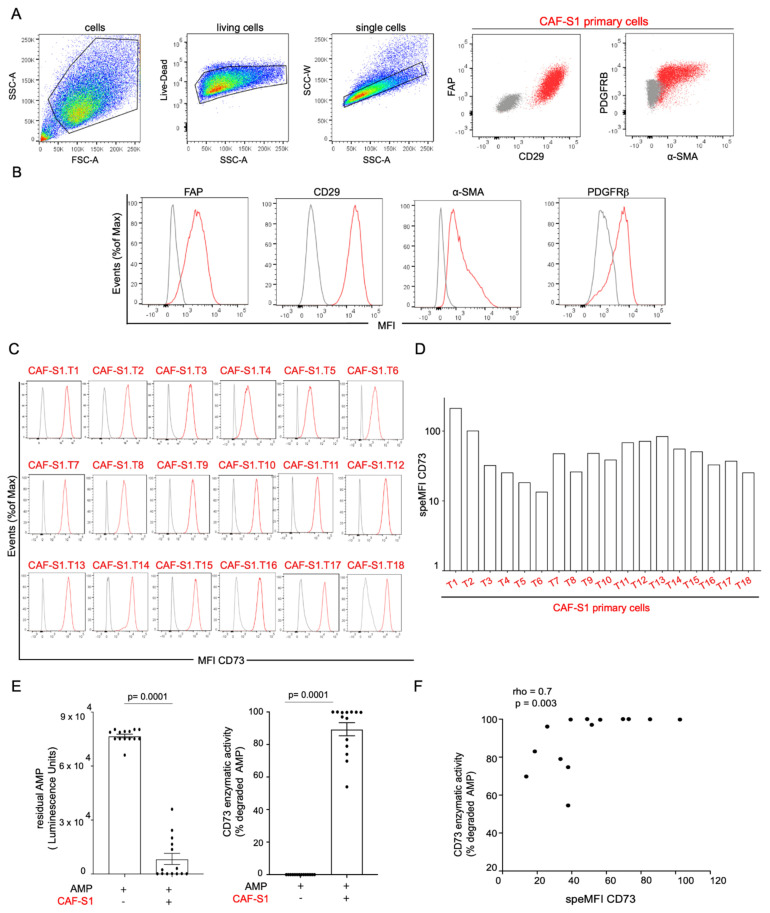
CD73 is enzymatically active at the surface of primary CAF-S1 cell lines. (**A**) Gating strategy from one representative patient to characterize human primary CAF cell lines by flow cytometry. Cells were gated to exclude dead (Live-Dead-) and doublets (SSC-A versus SSC-W) and analyzed with FAP, CD29, α-SMA and PDGFRβ CAF markers. Representative flow cytometry plots from CAF markers are shown (grey: isotype control; red: staining). (**B**) Mean fluorescence intensities (MFI) of four CAF markers (FAP, CD29, PDGFRβ, α-SMA) (red) compared to isotype control (grey) to confirm the CAF-S1 phenotype, as defined in [19,20,26], i.e. positive for all the tested markers. Cell counts are normalized, as percentages of maximal number of cells (% of Max). (**C**) Same as (**B**) assessing CD73 protein level (red) compared to isotype control (grey) at the surface of 18 human CAF-S1 primary cell lines (referred to as T1 to T18) from 18 BC patients. (**D**) Corresponding quantification of the CD73 surface protein levels in CAF-S1 cell lines. Data are plotted as specific median fluorescence intensity (speMFI) calculated as follow: speMFI = CD73 MFI/isotype control MFI. (**E**) Left: Residual exogenous AMP levels in supernatants of primary CAF-S1 cell lines. CAF-S1 were incubated with 50 µM AMP and supernatants collected after 1h to measure residual AMP with AMP-Glo^TM^ luminescence assay. Residual AMP = (relative luminescence unit in CAF-S1/relative luminescence unit of AMP alone). Each dot represents residual AMP in one primary CAF-S1 cell line (*n* = 14). Right: CD73 enzymatic activity assessed by the percentage (%) of degraded AMP in CAF-S1 calculated as follow: % degraded AMP = 100 − % residual AMP. For the two plots, data are shown as median ± SEM (*n* = 14). *p*-values from Wilcoxon test. (**F**) Correlation between CD73 enzymatic activity (assessed by % of degraded AMP) and CD73 surface protein level (expressed as CD73 SpeMFI) in CAF-S1 primary cell lines from BC (*n* = 18). Correlation coefficient and *p*-value are based on Spearman’s test. LumA: Luminal A; LumB: Luminal B; HER2: human epidermal growth factor receptor 2; TN: triple negative; CAF: cancer-associated fibroblast, NA: not available.

**Figure 3 cancers-13-05878-f003:**
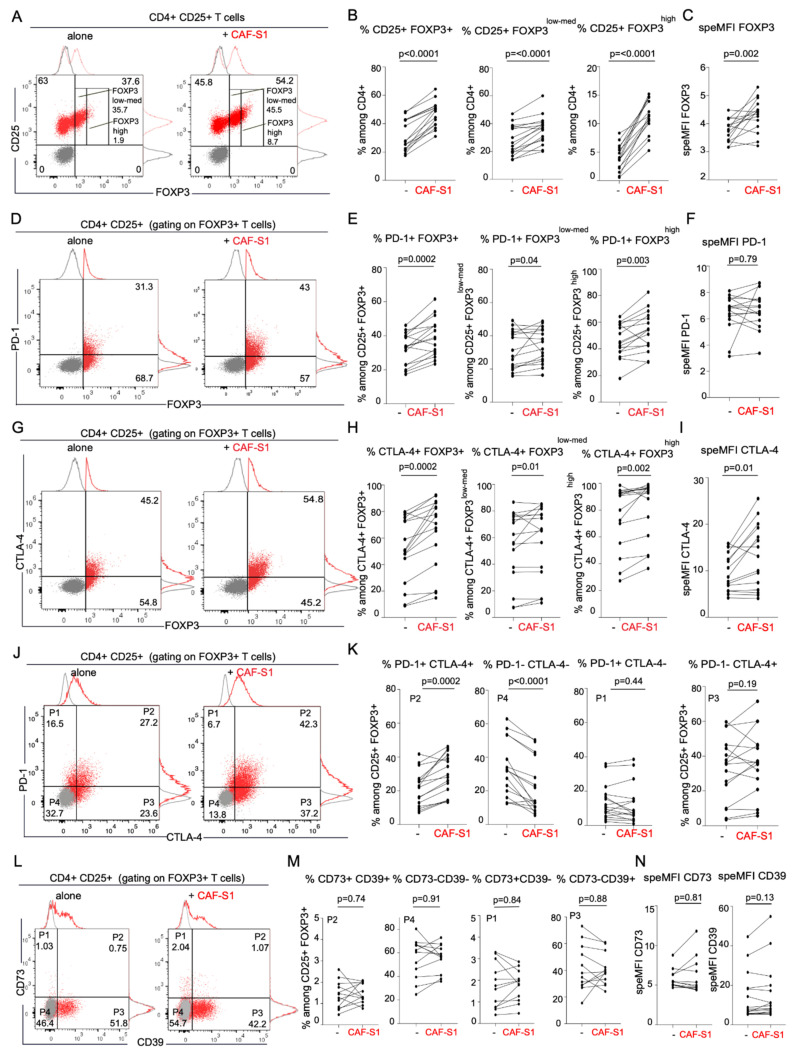
CAF-S1 promote the increase of CD25+ FOXP3+ PD-1+ CTLA-4+ Tregs. (**A**) Flow cytometry plots showing the proportion of CD25+ FOXP3+/- cells among CD4+ CD25+ T lymphocytes either alone (**left**) or in presence of CAF-S1 (**right**). (**B**) Percentages (%) of CD25+ FOXP3+ (**left**), CD25+ FOXP3^low-med^ (**middle**) and CD25+ FOXP3^high^ (**right**) T lymphocytes among CD4+ CD25+ T cells alone or upon co-culture with CAF-S1. Each dot represents one donor (*n* = 16). Five CAF-S1 independent primary cell lines tested. *p*-values from paired *t*-test. (**C**) FOXP3 specific MFI (speMFI = FOXP3 MFI/isotype control MFI) in CD4+ CD25+ FOXP3+ T cells alone or upon co-culture with CAF-S1. Each dot represents one donor (*n* = 16). Five CAF-S1 independent primary cell lines tested. *p*-values from paired *t*-test. (**D**) Same as in (**A**) for PD-1+ cells among CD4+ CD25+ FOXP3+ T lymphocytes either alone or in presence of CAF-S1. (**E**) Same as in (**B**) evaluating % of PD-1+ among CD4+ CD25+ FOXP3+ (Total) (**left**), FOXP3^low-med^ (**middle**) and FOXP3^high^ (**right**) either alone or upon co-culture with CAF-S1 (*n* = 16). *p*-values from paired *t*-test for % of PD-1+ among CD4+ FOXP3+ and FOXP3^high^. *p*-values from Wilcoxon test for FOXP3^low-med^ (**F**) Same as in (**C**) for PD-1 SpeMFI in CD4+ CD25+ FOXP3+ PD-1+ T cells alone or upon co-culture with CAF-S1 (*n* = 16). Five CAF-S1 independent primary cell lines tested. *p*-values from Wilcoxon test. (**G**) Same as in (**D**) for CTLA-4+ among CD4+ CD25+ FOXP3+ T lymphocytes either alone or in presence of CAF-S1. (**H**) Same as in (**E**) evaluating % of CTLA-4+ among CD25+ FOXP3+ (Total) (**left**), FOXP3^low-med^ (**middle**) and FOXP3^high^ (**right**) either alone or upon co-culture with CAF-S1 (*n* = 16). Five CAF-S1 independent primary cell lines tested. *p*-values from Wilcoxon test. (**I**) Same as in (**F**) for CTLA-4 SpeMFI in CD4+ CD25+ FOXP3+ CTLA-4+ T cells alone or upon co-culture with CAF-S1 (*n* = 16). *p*-values from Wilcoxon test. (**J**) Same as (**A**,**D**,**G**) for PD-1+ CTLA-4- (P1), PD1+ CTLA4+ (P2), PD1- CTLA-4+ (P3) and PD1- CTLA-4- (P4) among CD25+ FOXP3+ T lymphocytes either alone or in presence of CAF-S1. (**K**) Same as (**B**,**E**,**H**) evaluating % of PD-1+ CTLA-4+, PD-1- CTLA-4-, PD-1+ CTLA-4- and PD-1- CTLA-4+ among CD4+ CD25+ FOXP3+ T lymphocytes alone or upon co-culture with CAF-S1 (*n* = 16). Five CAF-S1 independent primary cell lines tested. *p*-values from Wilcoxon test. (**L**) Same as (**J**) for CD73 and CD39 in CD4+ CD25+ FOXP3+ T lymphocytes alone or in presence of CAF-S1. (**M**) Same as (**K**) evaluating % of CD73+ CD39+, CD73- CD39-, CD73+ CD39- and CD73- CD39+ among CD4+ CD25+ FOXP3+ T lymphocytes alone or upon co-culture with CAF-S1 (*n* = 12). 4 CAF-S1 independent primary cell lines tested. *p*-values from paired *t*-test (**N**) Same as (**F**,**I**) for CD73 SpeMFI (**left**) and speMFI CD39 in CD4+ CD25+ FOXP3+ CD39+ T cells or CD4+ CD25+ FOXP3+ CD73+ T cells alone or upon co-culture with CAF-S1 (*n* = 12). 4 CAF-S1 independent cell lines tested for CD73 and 5 for CD39. *p*-values from Wilcoxon test.

**Figure 4 cancers-13-05878-f004:**
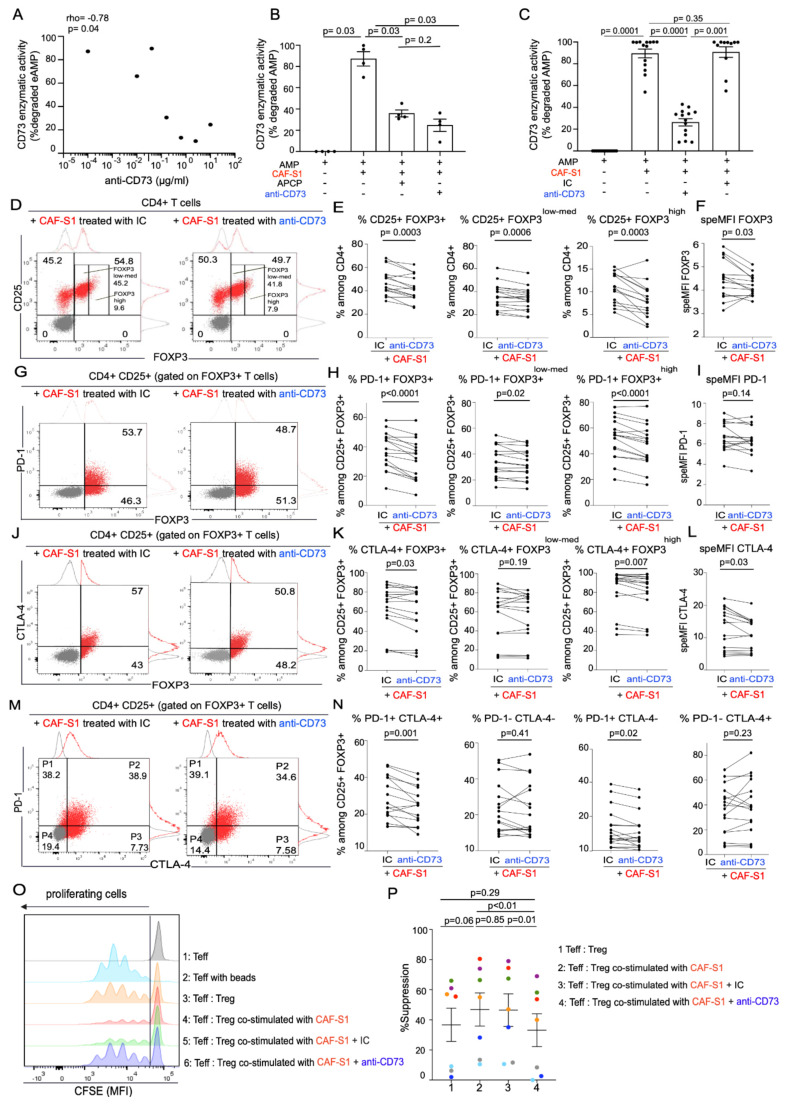
Anti-CD73 antibody impairs CAF-S1 immunosuppressive activity. (**A**) Inverse correlation between CD73 enzymatic activity (assessed by % of degraded AMP = 100 − % residual AMP) in CAF-S1 and the different concentrations of anti-CD73 antibody tested, as indicated. Correlation coefficient and *p*-value from Spearman’s test (*n* = 4 independent experiments) (**B**) CD73 enzymatic activity in CAF-S1 in presence of the pharmacological inhibitor APCP (100 µM) and compared to monoclonal anti-CD73 antibody (at 10 µg/mL, as defined in (**A**)). % degraded AMP = (100 − % residual AMP) and % residual AMP = 100 × (relative luminescence unit in CAF-S1 (untreated, treated with APCP or anti-CD73 Ab)/relative luminescence unit of AMP alone). Each dot represents data from one primary CAF-S1 cell line (*n* = 4). Data are mean ± SEM. *p*-values from Mann–Whitney test (**C**) CD73 enzymatic activity in CAF-S1 in presence of isotype control (IC) and monoclonal anti-CD73 antibody (at 10 µg/mL, as defined in (**A**)). % degraded AMP = (100 − % residual AMP) and % residual AMP = 100 × (relative luminescence unit in CAF-S1 (untreated, treated with anti-CD73 Ab or with IC)/relative luminescence unit of AMP alone). Each dot represents data from one primary CAF-S1 cell line (*n* = 14). Data are mean ± SEM. *p*-values from Wilcoxon test. (**D**) Flow cytometry plots representing the proportion of CD25+ FOXP3+/- cells among CD4+ CD25+ T lymphocytes in presence of CAF-S1 pre-treated with IC (**left**) or with anti-CD73 antibody (**right**). Treatment of CAF-S1 with anti-CD73 Ab or IC antibodies (10 µg/mL) was performed for 24h before CAF-S1 co-culture with CD4+ CD25+ T lymphocytes. (**E**) Percentages (%) of CD25+ FOXP3+ (**left**), CD25+ FOXP3^low-med^ (**middle**) and CD25+ FOXP3^high^ (**right**) among CD4+ CD24+ T lymphocytes upon co-culture with CAF-S1 pre-treated with IC or with anti-CD73 Ab. Each dot represents one donor (*n* = 16). Five CAF-S1 independent primary cell lines tested. *p*-values from paired *t*-test. (**F**) FOXP3 speMFI among CD4+ CD25+ FOXP3+ T lymphocytes upon co-culture with CAF-S1 pre-treated with IC or with anti-CD73 Ab. Each dot represents one donor (*n* = 16). Five CAF-S1 independent primary cell lines tested. *p*-values from Wilcoxon test. (**G**) Same as (**D**) for PD-1+ among CD4+ CD25+ FOXP3+ T lymphocytes. (**H**) Same as (**E**) evaluating % of PD-1+ among CD4+ CD25+ FOXP3+ (**left**), CD25+ FOXP3^low-med^ (**middle**) and CD25+ FOXP3^high^ (**right**) upon co-culture with CAF-S1 pre-treated with IC or with anti-CD73 Ab (*n* = 16). *p*-values from paired *t*-test for % of PD-1+ among CD4+ CD25+ FOXP3+ and CD25+ FOXP3^high^. *p*-values from Wilcoxon test for % of PD-1+ among FOXP3^low-med^ (**I**) Same as in (**F**) for PD-1 SpeMFI in CD4+ CD25+ FOXP3+ PD-1+ T cells. Each dot represents one donor (*n* = 16). Five CAF-S1 independent primary cell lines tested. *p*-values from Wilcoxon test. (**J**,**K**,**L**) Same as (**G**,**H**,**I**) evaluating % and SpeMFI of CTLA-4+ among CD4+ CD25+ FOXP3+ CTLA-4+ T lymphocytes (*n* = 16). Five CAF-S1 independent primary cell lines tested. *p*-values Wilcoxon test. (**M**) Same as in (**D**,**G**,**J**) for PD-1+ CTLA-4- (P1), PD-1+ CTLA4+ (P2), PD-1- CTLA-4+ (P3) and PD-1- CTLA-4- (P4) among CD4+ CD25+ FOXP3+ T lymphocytes. (**N**) Same as in (**E**,**H**,**K**) for PD-1+ CTLA-4+, PD-1- CTLA-4-, PD-1+ CTLA-4- and PD-1- CTLA-4+ T lymphocytes. Each dot represents one donor (*n* = 16). Five CAF-S1 independent primary cell lines tested. *p*-values from Wilcoxon test. (**O**) Representative CSFE fluorescent intensities quantifying CD4+ effector T cells (Teff) proliferation. Teff were incubated in presence of CD3/CD28 beads and Tregs (CD4+ CD25^med^ CD128^low^ CD45RA^low^) (Treg: Teff ratio of 2:1), Tregs having been either pre-incubated with untreated CAF-S1 or with CAF-S1 pre-treated with IC (green) or with anti-CD73 Ab (blue). (**P**) Quantification of the percentage (%) of suppression of Teff proliferation by Tregs, calculated as follow: ((log2 (y) of Teff alone − log2(y) Teff + Treg)/(log2 (y) Teff alone)) × 100, where y corresponds to CFSE MFI in the whole population divided by CFSE MFI in non-proliferating cells. Data are means ± SEM. Each dot represents one donor (*n* = 5). Three CAF-S1 independent primary cell lines tested. *p*-values from paired *t*-test.

**Table 1 cancers-13-05878-t001:** Description of the prospective and retrospective cohorts used in the study. Retrospective cohort related to Figure 1; Prospective cohort related to Figures 2–4.

Retrospective Cohort							Prospective Cohort
Total number of patients		215					26
Gender	Female	215					25 (96%)
	Male	0					1 (4%)
Date of Inclusion		2004–2012					2013–2020
Mean follow-up (years)		7.7					1.5
Mean age at diagnosis (years)		55 (22 min–87 max)					53 (29 min–76 max)
	I	41 (19%)					3 (11%)
Histological grade	II III	67 (31%) 105 (49%)					9 (35%) 13 (50%)
	NA	2 (1%)					1 (4%)
	pT0	0					3 (11%)
	pT1	146 (68%)					10 (39%)
Pathological tumor	pT2	62 (29%)					11 (42%)
size (pT)	pT3	4 (1.5%)					1 (4%)
	pT4	3 (1.5%)					0
	NA	0					1 (4%)
Pathological Lymph node status (pN)	Negative Positive NA	127 (59%) 86 (40%) 2 (1%)					21 (81%) 4 (15%) 1 (4%)
Metastatic status	Negative Positive NA	203 (94%) 3 (2%) 9 (4%)					25 (96%) 0 1 (4%)
Mean tumor size (mm)		19					23.7
	Lum A	96 (45%)					9 (35%)
Breast cancer subtype	Lum B HER2	- 54 (25%)					3 (11.5%) 3 (11.5%)
	TN	65 (30%)					11 (42%)
	CAF-S1	32 (15%)					-
	CAF-S2	38 (18%)					-
CAF enrichment	CAF-S3	3 (1%)					-
	CAF-S4	111 (52%)					-
	NA	31 (14%)					-
		CAF-S1	CAF-S2	CAF-S3	CAF-S4	NA	
CAF enrichment in breast cancer subtypes	LumA HER2	6 (6%)	34 (35%)	1 (1%)	40 (42%)	15 (16%)	- -
		9 (17%)	2 (3%)	1 (2%)	33 (61%)	9 (17%)	
	TN	16	2	1	38	8	-
		(25%)	(3%)	(1%)	(59%)	(12%)	
	Yes	85 (39%)					11 (42%)
Hormonotherapy	No	126 (59%)					0
	NA	4 (2%)					15 (58%)
	Yes	204 (95%)					21 (81%)
Radiotherapy	No	0					4 (15%)
	NA	11 (5%)					1 (4%)
	Yes	141 (66%)					19 (73%)
Chemotherapy	No	0					0
	NA	74 (34%)					7 (27%)
	Yes	45 (21%)					2 (8%)
Targeted therapy	No	0					0
	NA	170 (79%)					24 (92%)

List of abbreviations: pT: Pathological tumor size; pN: Pathological Lymph node status: mm: millimeters; Breast cancer subtypes: Lum A/B: Luminal A/B; HER2: Human epidermal growth factor receptor 2; TN: Triple Negative; NA: Not available.

## Data Availability

The data presented in this study are available on request from the corresponding authors.

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
