# Peer review of "CD73-Mediated Immunosuppression Is Linked to a Specific Fibroblast Population That Paves the Way for New Therapy in Breast Cancer"

_cancers, 2021, doi:10.3390/cancers13235878_

Round 1

Reviewer 1 Report

This manuscript address an important issue related to breast cancer immune suppression. It focuses on the role of CD73, expressed on some cancer-associated fibroblasts as a leading factor promoting T-reg cells. However, elevated CD73 levels in the tumor tissue of several cancer types, including breast has been reported. In addition, CD73 neutralization therapy is being tested in clinical trials. CD73 activity in cancer-associated fibroblasts (CAFs) has been also recently reported.

The ex-vivo studies performed by the authors are very significant. However, modulation of CD73 needs to be assessed further. Is cell confluence, “Breast cancer cells”/fibroblasts interactions, potential signaling in TME…, play a role on CD73 expression?

Author Response

We are really grateful to the Reviewer for his/her positive assessment of our work. We thank the Reviewer for considering our paper as addressing an important issue in breast cancer and defining CD73 as a key stromal factor involved in immunosuppression.

Compared to previous studies, including our own, we would like to emphasize that our current manuscript quantifies CD73 histological scores both in stromal and in epithelial compartments. Moreover, these quantifications are performed in a high number of breast cancer patients, including the three molecular subtypes. This is also the first time that CD73 immunosuppressive function in human breast cancer is specifically mediated by the stroma, and not by the epithelium. Thus, we provide here one of the first in-depth and highly-resolutive study on CD73 protein function in breast cancer. We thank the Reviewer for his/her advice and we now underline this point in a more detailed manner in the Discussion, p18.

Regarding CD73 expression, we observed that CD73 is highly detected -both at mRNA and protein levels- in CAF-S1, compared to the other CAF populations. CD73 expression is well-known to be regulated by TGFb-signaling pathway (Chen, Nat Commun, 2019). Interestingly, we previously highlighted the role of TGFb signaling in CAF-S1 functions and its reciprocal crosstalk with cancer cells (Pelon et al., Nat Commun, 2020), suggesting TGFb might be a key regulator of CD73 expression in CAF-S1. Moreover, it has been shown that the crosstalk between CAF and T lymphocytes plays an important role in the regulation of stromal CD73 expression. Indeed, ex-vivo study with CAF isolated from lung cancer showed how T cells up-regulates CD73 on CAF in lung cancer through INFγ secretion (O’Connor, Oncoimmunology, 2021). As recommended by the Reviewer we now discuss these different points in the new version of the  text and added the corresponding references, p19.

Reviewer 2 Report

A manuscript that could have been a significant contribution to understanding breast cancer's cellular biology is lost in translation. The hypothesis is valid, experimental set up all logical with the expected outcome; tables and figures are clear with essential details. The short discussion justifies the conclusion derived from the studies and gives credibility and a necessary message and effect. All the in-vitro studies support this paper's essence for future clinical trials and contribute to managing patients with breast cancer. The manuscript is highly detailed, and this is the pitfall of this article, too complicated and complex. The methods are described in unnecessary details that make the reading of the manuscript difficult and dull; the readers will lose interest to continue reading. The authors must shorten methods with proper referencing. The article as it stands now is tedious, confusing; too many repetitive details will destroy the main message and outcome of the studies. In short, this article is not in publishable form unless it's revised completely to remove repetitions and unnecessary details, stick to the fact of the studies. Authors should provide an abbreviation list. Pay careful attention to the typographical errors in table one and other places in the manuscript. 

Author Response

We thank the Reviewer for the positive evaluation of our manuscript and for his/her interesting suggestions. We have now considered his/her comments and provided the corresponding modifications. As recommended by the Reviewer, we have now simplified the methods and eliminated unnecessary details and repetitions, when this was the case. Thanks to this suggestion, the Methods section (p7) has been reduced almost by half, which greatly helps in its understanding. We also cited the adequate references. All modifications are in apparent for their better visualization. We also provided the significance of all abbreviations all along the text and a list of abbreviations recapitulating their meaning, as requested (p2-3). Finally, we have corrected the typographical errors in Table 1, as recommended. We thank the Reviewer for his/her critical reading that improves our manuscript.

Reviewer 3 Report

The manuscript by Magagna et al. is well-written,  well-conducted and conclusions are supported by the data. It shows additional results on CD73 enriched CAFs contributing to immunosuppression. A previous manuscript from the same group was published before (2018) on Cancer Cell.  

 My main concerns/comments are:

1) The methodology is rich, well documented, and well described, facilitating future studies based on this paper due to its high reproducibility. However, there are repetitive sentences that your text could benefit with its removal or even re-structure. I have suggested a few changes and have some technical questions about this segment.

2) Please, provide abbreviations in Table 1’s subtitle for a better understanding of the dataset.

3) Also on Table 1, the % of patients NA for Hormonotherapy is 577%. Was it 57.7%? Please, correct.

4) The abbreviation for room temperature (RT) was not defined prior to its usage on page 5:

“Blocking of unspecific binding was performed using a Dako blocking serum- 155 free solution ready-to-use (Dako, #X0909) for 10 min at room temperature. TMA sections 156 were incubated with rabbit anti-human CD73 primary antibody (Sigma, #HPA017357, di- 157 lution 1:600) previously tested for its specificity or Rabbit IgG, polyclonal isotype control 158 (abcam, #ab171870, dilution 1:1250) for 1h at RT.”

5) Why focus on CAFs population in BC prior to treatment? It could be interesting to verify the CAF population of BC tumors after treatment in recurrence cases, checking it the treatment failure has something to do with CAFs population.

6) About CAF-S1 senescence phenotype accessed in later passages. The paper could benefit from a supplementary figure with Beta-galactosidase test and/or the contrast phase morphology of CAFs prior to the 10th passage.

7) In page 5, CAFs culture is well explained and detailed. However, define “(…) DMEM supplemented with 10% heat-inactivated Fetal Bovine Serum (FBS) with streptomycin and penicillin (100 U/ml) (…)” just as Supplemented DMEM. The methods section is long and could benefit from this simplification. Also, the abbreviations, such as Fetal Bovine Serum (FBS), should be defined once.

8) Please, standardize if the number of cells will be represented in scientific notation or not.

“CAF were suspended in PBS supplemented with 1% heat-inactivated human serum and 2 mM EDTA at the 500 000 cells in 50 μl and were moved in 96 well-plate V-bottom (Greiner, #651101) for the staining. (page 6)

“CAF were suspended in PBS supplemented with 1% human serum and 2 mM EDTA at the following concentration of 5 × 105 cells in 50 μl and transferred in 96 well-plate V-bottom (Greiner, 265 #651101) for the staining.” (page 7)

9) Page 5, line 182, please cite the previous studies.

10) Why DAPI staining, but not one LIVE/DEAD fixable dye, was used in this experiment (2.6. Measurement of CD73 protein levels in CAF-S1 by flow cytometry)?

“Dead cells were excluded based on their positive staining for DAPI.” (page 7)

11) Please, provide a Statistics subsection with all the tests made and the software used

Results

12) Overall, the results section is very well written, but lacks details about the quantitative aspects of the data. It would be interesting to include mean/median values and their errors in the description of the results. A contrast microscopy image of CAF-S1 with and without the T lymphocytes would add a morphological value to the results.

“We first observed that CD73 protein level was heterogeneous from one tumor to another in the different BC subtypes (Figure 1A).” (page 11)

13) How heterogeneous? Was the CD73 expression more diffuse in one subtype than in the other? Was it more delimited? Please, provide a better description for Figure 1A.

Also, the figure 1A inserts don’t present a higher magnification than the original image, and it is difficult to observe CD73 staining, especially on stromal fibroblasts.

14) Figure 1B would benefit from colors because it is difficult to see the median/mean line among the dots. Dots in different color among the analyzed groups and the median/mean line could be black (same as shown on Fig. 1d).

15) Please, provide the list of abbreviations used in Figure 1, preferentially in the subtitle.

“We confirmed that CD73 protein was detected at the surface of all CAF-S1 primary cell lines but at variable levels, as observed in stroma of BC patients (Figure 2C, D).” (page 13)

16) Please, provide descriptive statistics (average/median, error, min, and max values etc) of the CD73 expression among the 18 samples. Include the MFI (MFI of marked/ MFI of control) of each histogram in Figure 2 could enrich the graphical representation of results.

“As reported in breast and ovarian cancers [19, 20, 26], we confirmed that 583 primary CAF-S1 cells co-cultured with CD4+ CD25+ T cells significantly increased the percentage of CD4+ CD25+ FOXP3+ T lymphocytes in each (Figure 3), with a strong impact on CD4+ CD25+ FOXP3high sub-population (Figure 3B, right).” (page 15)

17) Increased how? Specify the fold-change of the % of T lymphocytes in the description of Figure 3 results.

18) The discussion presents well written, have up-to-date references, but the paragraphs are too extensive. I suggest that the authors revise the text and separate the paragraphs based on core ideas/concepts based on the subtopics of the results section.

For example:

The first paragraph could be divided in two, starting in line 720 “A meta-analysis has shown that FOXP3+ Tregs have a significant negative effect on overall survival (OS) in BC patients, and are associated with estrogen and progesterone receptor status, as well as with lymph node metastases [48-50].”

I also suggest breaking the second paragraph in two paragraphs starting in line 752 “In line 752 with this, low CD73 expression and high global infiltration of CD45+ immune cells exhibit 753 additive prognostic values and are indicative of a good clinical outcome in TN BC patients 754 with invaded lymph nodes [61].”.

19) The font style is different between lines 742 and 744.

The following papers can enhance the discussion, please cite:

  • 10.1080/2162402X.2021.1940675
  • https://doi.org/10.1016/j.coph.2020.07.001

Author Response

We first would like to thank the Reviewer for his/her careful reading and positive assessment of our manuscript. We follow all the recommendations provided by the Reviewer 3, which helped us a lot to improve our manuscript. We thus warmly thank the Reviewer for all these suggestions. All modifications are in apparent in the new version of the text for their better visualization.

Main comments

1) We have now shortened the Methods section. As recommended by the Reviewer, we have now simplified the methods and eliminated unnecessary details and repetitions, when this was the case (p p7-13).

2) As requested, we have now provided a list of abbreviations in Table 1’s subtitle for a better understanding of the dataset.

3) We are sorry for this typographical error, which has now been corrected in Table 1, as requested. We have also reduced the numbers of decimals of the percentages to simplify Table 1 and to help the reader.

4) As requested, the abbreviation for room temperature (RT) is now defined before use p8 in the new version of the text and has been corrected accordingly -for appropriate use- all along the text.

5) We do agree with the Reviewer that considering the impact of treatment on CAF populations is interesting. In line with this, we are currently investigating this question and have indeed observed a strong and complex effect of chemotherapy on CAF subpopulations. Description of the impact of treatment on CAF subpopulation is a complete study in itself. We hope that the Reviewer will agree that this effect is beyond the scope of the current study on the impact of stromal CD73 in immunosuppression in breast cancer.

6) As recommended by the Reviewer, we have now included phase contrast images of 3 representative primary CAF-S1 cell lines isolated from 3 different patients prior to the 10th passage. These data have been inserted in Supplementary Figure S2A.

7) We thank the Reviewer for this suggestion of simplification. We have defined the “Supplemented DMEM” one (p8-9), and next used this nomenclature all along the text, as recommended by the Reviewer. The Methods section benefits from this specific simplification. We have also simplified the Methods and eliminated unnecessary details and repetitions. Thanks to these modifications, the Methods section (p7) has been reduced almost by half, which greatly helps in its understanding.

In addition, as requested, the Fetal Bovine Serum (FBS) has been defined once (end of p8). To help in the understanding of our manuscript, we also now provide a list of abbreviations recapitulating their meaning (p2-3).

8) As requested, we have now standardized and represented the number of cells in scientific notation, all along the Methods section.

9) As requested, we have modified the sentence for better highlighting the cited references, p7.

To determine the CAF-subset enrichment in BC patients of each subtype, the staining … by immunohistochemistry (IHC) in LumA, HER2, and TN BC subtypes and the histological scores (Hscores) quantified, as described in [19, 28].”

10) In this specific section, we used DAPI instead of Live/Dead because only surface staining was performed, while in previous section, this was both surface and intracellular staining.

11) We are sorry for the missing paragraph in the first version of our manuscript. We thank the Reviewer for this suggestion, and we now provide a “Statistical analysis” subsection in the Methods part, p13.

Results

12) We have included the values in the description of the Results, as recommended. We also now provide phase contrast image of CD4+ CD25+ T lymphocytes with and without CAF-S1 in the new Supplementary Figure 2B.

13) As recommended, we have now re-written the description of these data (p14), to clarify the description of the heterogeneity of CD73 staining in tumors and in stromal and epithelial compartments.

CD73 protein level and pattern were assessed by immunohistochemistry (IHC) staining on these BC samples. We first observed that CD73 protein level was heterogeneous in the different BC subtypes, as well as from one tumor to another within the same BC subtype (Figure 1A)…… However, different CD73 protein levels were observed not only between different BC subtypes, but also between cellular compartments. Some tumors showed a strong staining in cancer cells (epithelial compartment) but a weak staining in CAF (stromal compartment), while other tumors exhibited high staining in CAF but low in cancer cells. We thus evaluated CD73 protein levels in each BC sample considering epithelial and stromal compartments separately (Figure 1B).” This text is now added p14 in the new version of the manuscript.

In addition, we have now included inserts with higher magnification of each photo in Figure 1A, as requested.

14) As recommended, we now use different colors among the analyzed groups in Figure 1B and the median/mean lines remain in black to better visualize them. We thank the Reviewer for this suggestion helping the reader to visualize the data.

15) We have now provided a list of abbreviations in the subtitle of the Table 1, as requested. As also mentioned above, we have also incorporated a list of abbreviations covering the entire paper, p2-3.

16) We thank the Reviewer for this suggestion. We have now provided in the text the statistics (mean, error, min, and max values) of the CD73 specific MFI among the 18 CAF-S1 primary cell lines isolated from human breast cancers. These values are now given p15. We have also inserted all corresponding flow cytometry histograms in the New Figure 2C, as requested.

17) As recommended, we have now included the fold changes of the percentages of the CD4+ CD25+ FOXP3+ and CD4+ CD25+ FOXP3high sub-population among CD4+ CD25+ T lymphocytes upon co-culture with CAF-S1 fibroblasts. These information are given in the new manuscript (p16).

18) As mentioned by the Reviewer, we have now revised the Discussion (p18-19) by separating the paragraphs into different subsections according to the subtopics.

19) The two references have been added, as recommended (p19). We thank the Reviewer for these suggestions.

Round 2

Reviewer 1 Report

The authors nicely put their discovery in the current context of research on the role of CD73 in cancer. I endorse the publication of revised manuscript.

Reviewer 2 Report

No further comments are required, the article is comprehensively revised and is in publishable form.